# Light-Trapping-Enhanced Photodetection in Ge/Si Quantum Dot Photodiodes Containing Microhole Arrays with Different Hole Depths

**DOI:** 10.3390/nano12172993

**Published:** 2022-08-30

**Authors:** Andrew I. Yakimov, Victor V. Kirienko, Dmitrii E. Utkin, Anatoly V. Dvurechenskii

**Affiliations:** 1Rzhanov Institute of Semiconductor Physics, Siberian Branch of the Russian Academy of Science, 630090 Novosibirsk, Russia; 2Physical Department, Novosibirsk State University, 630090 Novosibirsk, Russia

**Keywords:** quantum dots, near-infrared photodetection, photon-trapping microstructures, telecom

## Abstract

Photodetection based on assemblies of quantum dots (QDs) is able to tie the advantages of both the conventional photodetector and unique electronic properties of zero-dimensional structures in an unprecedented way. However, the biggest drawback of QDs is the small absorbance of infrared radiation due to the low density of the states coupled to the dots. In this paper, we report on the Ge/Si QD pin photodiodes integrated with photon-trapping hole array structures of various thicknesses. The aim of this study was to search for the hole array thickness that provided the maximum optical response of the light-trapping Ge/Si QD detectors. With this purpose, the embedded hole arrays were etched to different depths ranging from 100 to 550 nm. By micropatterning Ge/Si QD photodiodes, we were able to redirect normal incident light laterally along the plane of the dots, therefore facilitating the optical conversion of the near-infrared photodetectors due to elongation of the effective absorption length. Compared with the conventional flat photodetector, the responsivity of all microstructured devices had a polarization-independent improvement in the 1.0–1.8-μm wavelength range. The maximum photocurrent enhancement factor (≈50× at 1.7 μm) was achieved when the thickness of the photon-trapping structure reached the depth of the buried QD layers.

## 1. Introduction

Uncooled Si quantum dot (QD) photodetectors (QDIPs) containing Ge QDs as an active absorbing element are capable of detecting optical radiation in the near-infrared (NIR) wavelength range (1.2–2.0 μm) [1,2,3,4] and can be used for development of efficient methods for transmitting information in television and telephone networks, the Internet, and optical computers. Despite the superior features of photodetectors with QDs, the QDIP’s currently achieved sensitivity to IR radiation is small, which results from the low density of the states associated with QDs and from the limited QD absorption thickness. To increase the absorption and hence the responsivity of low-absorbance devices, miscellaneous structures have been incorporated into photodetertors [5]. Plasmonic metasurfaces [6,7,8,9] and photon-trapping structures [10,11,12,13,14,15,16,17] have been demonstrated to be effective in reinforcement of photoresponses. The disadvantages of metallic metasurfaces that allow conversion of the incident electromagnetic radiation into the surface plasmons are high ohmic losses in the metal [18] and a small penetration depth of the plasmon field to the semiconductor, particularly for short wavelengths. Another efficient solution is to ensure a longer path of light through the detector using all-dielectric photon-trapping structures, which offer a low-loss alternative to plasmon elements [19,20,21,22,23,24]. Micro- and nanohole arrays on the detector’s surface enable light trapping with the generation of the lateral modes. The propagation of photons in the larger lateral dimension elongates the effective absorption length and enhances photon–material interaction, benefitting the increased optical absorption.

In a recent study, we reported on the vertical Ge/Si QD pin photodiodes with self-assembled Ge quantum dots grown on a silicon-on-insulator (SOI) substrate for photodetection in the NIR telecommunication wavelength range [25]. The photon-trapping 2D hole arrays were introduced into QDIPs to convert the incident electromagnetic radiation to the lateral collective modes. Compared with the non-photon-trapping counterpart, the photon-trapping QDIP exhibited an ≈30× responsivity enhancement at a wavelength of around 1.6 μm. A typical QDIP contains multiple stacks of buried QD layers as well as heavily doped contact regions at the top and bottom (Figure 1a). The microscale holes are etched through the top contact and absorbing layers for promoting lateral propogating modes. By changing the hole depth, one can tune the vertical spreading of the laterally traveling modes across detector. Using a light-trapping structure with holes passing through the QD absorber is beneficial to optical field confinement within the QD active region but at the expense of a decreased number of QDs, which results in less absorption. Thus, there is a trade-off between the amount of absorbing semiconductors remaining after etching and the enhanced light intensity in the region of the QDs. This work is a continuation of the research outlined in a previous paper [25]. Here, to address the trade-off, we analyzed and compared light-trapping enhanced Ge/Si QDIPs with different hole depths. By incorporating a circular hole array structure in square lattice with an optimal hole depth to the heterostructure, the photoresponse in the NIR wavelength range can be significantly enhanced. It has been demonstrated that the maximum photocurrent enhancement (50× at 1.7 μm) is achieved when the thickness of the photon-trapping structure reaches the depth of the buried QD layers. The dependence of the photoresponse on the polarization of the incident radiation has also been examined.

## 2. Materials and Methods

The light-trapping QDIPs were typical vertical mesa-type photodiodes fabricated by a solid source molecular beam epitaxy (MBE) using a Riber SIVA-21 system. In the experiments, a (001)-oriented SOI wafer with a 200-nm top silicon film and 2000-nm buried silicon oxide was used as the substrate. The sample geometry is shown in Figure 1a. The insulator platform plays the role of a bottom reflector and can enhance the absorption of normal incident light. Meanwhile, the employment of an SOI substrate makes possible the vertical Fabry–Pérot resonances between the SOI substrate and the top surface. The preparation started with a 20-nm-thick undoped Si layer. Then, the boron-doped ($5\times 10^{18}$ cm ${}^{-3}$) p-type Si bottom contact layer with a 500 nm thickness was grown at 600 °C. The active region of the QDIPs was composed of 10 stacks of Ge quantum dots separated by 10-nm Si barriers and was sandwiched in between the 200-nm-thick intrinsic buffer and cap Si layers fabricated at 500 °C and 400 °C, respectively. Each Ge QD layer consisted of a nominal Ge thickness of about 0.9 nm and formed at 250 °C at a rate of 0.04 nm/s by self-assembling in the Stranski–Krastanov growth mode. The QDs had a hut-like shape with a lateral size of 9.4 ± 3.2 nm and a height of about 1 nm. The surface density of the QDs was $5.2\times 10^{11}$ cm ${}^{-2}$. The average Ge content of 88% and the in-plane strain $\varepsilon_{\left|\right|}=-0.036$ in the Ge nanoclusters were determined from Raman scattering experiments using an approach developed in [26]. Finally, an Sb-doped 50-nm-thick n-Si top contact layer (∼10${}^{19}$ cm ${}^{-3}$) was grown at 400 °C to form a pin diode structure. Note that the QD active region with a thickness of 90 nm was located at a distance of $h_{\mathrm{QD}}=260$ nm below the Si–air surface (Figure 2). Details on the structure growth can be found in [4,25,27].

After the MBE growth, the wafers were processed into 700-μm diameter vertical pin QDIPs (Figure 1b). The hole arrays were etched to different depths ranging from 100 to 550 nm with a cylindrical shape profile, serving as the photon-trapping structures. It has been shown previously that Ge/Si QDIPs cover the NIR spectral range up to 1.6–1.8 μm [4,25]. In the present experiments, we fixed the hole diameter at 1300 nm and lattice periodicity at 1700 nm. The surface morphology was controlled by scanning electron microscopy (SEM) and atomic force microscopy (AFM). Some representative images are shown in Figure 1c–f. The air holes were produced by means of the reactive ion etching of the Si and Ge layers through a metallic template. The template was a 30-nm-thick perforated Cr film formed on the surface of the heterostructure by electron beam lithography, deposition of the metal in av acuum, and the subsequent lift-off process. The hole arrays had the square lattice symmetry with hole depths *h* = 100, 200, 270, 315, 380, and 550 nm. A reference QDIP without any texture pit was also fabricated for comparison of the device performance. The top and bottom ohmic contacts were made using Au evaporated onto the sample surface and annealing at 350 °C for 5 min in an Ar atmosphere. An optical image of the fabricated device is displayed in Figure 1b. The photocurrent measurements were performed at room temperature. The normal-incidence photoresponse was obtained using a Bruker Vertex 70 Fourier spectrometer with a spectral resolution of 30 cm ${}^{-1}$ along with an SR570 low-noise current preamplifier. A halogen lamp was used as a source of radiation. The photocurrent spectra were calibrated with a deuterated L-alanine-doped triglycine sulfate detector. The responsivity was determined by illuminating the samples with InGaAsP LEDs (Roithner Laser Technik), which were emitting at 1.3 and 1.55 μm. The room temperature dark current was tested as a function of a bias voltage between −2 V and +2 V by a Keithley 6430 Sub-Femtoamp Remote SourceMeter.

## 3. Results and Discussion

The reflection spectra of the devices with and without photon-trapping hole arrays are plotted in Figure 3. The incident depolarized the IR light-illuminated detectors from their front sides. The spectra were normalized by the reflection intensity spectrum of the Au flat mirror. As illustrated in Figure 3, large oscillations were observed on the spectra of the reference QDIP and the light-trapping device with shallow holes (*h* = 100 nm), providing evidence for the strong vertical resonances between the SOI substrate and top surface. The spectra resembled each other, and a slight difference in the peak position was caused by different effective refractive indices. The light in the detector’s active region experienced a strong reflection at the SiO${}_{2}$/Si and air/Si interfaces, creating a resonant cavity. The point of Figure 3 is to demonstrate that when the holes were deeper than 100 nm, the light-trapping QDIPs had lower reflection (10–20%) and much weaker vertical resonance, suggesting a successful conversion of an initial incident vertical plane wave to an ensemble of lateral collective modes, which are less dependent on constructive or destructive optical interference [28].

Figure 4 depicts the measured responsivity spectra of the detectors with flat and textured surfaces. A broad photoresponse in the NIR spectral range up to 1.8 μm was due to interband transitions between the electron states in the conduction band and the hole states bound inside Ge QDs [4]. Similar to the reflection spectra, the photocurrent characteristics of the control QDIP consisted of sharp resonances caused by the vertical cavity effect. With the incorporation of a light-trapping microstructure, the vertical resonances became less pronounced due to generation of the lateral guided modes. To test the polarization dependence of the spectral response, we measured the responsivity characteristics of the QDIPs under a linear polarized light with different polarization angles. We used a KRS-5 polarizer which was placed in the path of the IR beam to select the desired beam polarization. As shown in Figure 4h, the photocurrent spectra did not change with the polarization angles due to the structure symmetry of the hole arrays.

To gain clear insights into the enhancement from the light-trapping structures, the enhancement factor was analyzed. To obtain the hole-induced photoresponse enhancement factor, the experimental responsivity spectra were normalized to the reference spectrum of the control sample with a flat surface. The photocurrent enhancement at 2.5 V is plotted in Figure 5a. The enhancement spectrum of the 270-nm hole device at a 0 V bias is also shown by the black broken line. We found that the photocurrent enhancement in the photovoltaic regime was small, and it increased with the increasing reverse bias and ceased to depend on the voltage as the reverse bias voltage reached ≈1 V. This observation can be explained by electrically controlled overlap between the depletion layer and the light-trapping region [29].

At bias voltages larger than 1 V, the detector response was greatly enhanced by embedding the photon-trapping structure, where the maximum enhancement rate was obtained at wavelengths of around 1.7 μm. Compared with the conventional flat QDIP, the responsivity of all microstructured devices had broadband (10 − 20)× improvement at the wavelength range from 1.0 to about 1.6 μm and enhancement of 50× at 1.7 μm for the QDIP with *h* = 270 nm. It is necessary to pay attention to the fact that the maximum enhancement occurred when λ≃p, where *p* is the periodicity of the light-trapping structure. Under this condition, a photonic crystal was formed. Photonic crystals allow slow Bloch modes with a group velocity of photons close to zero, resulting in a strong enhancement in light–matter interaction [30,31,32].

In Figure 5b, we plot the enhancement factor integrated over the wavelength range from 1.6 to 1.8 μm. An important result is that the better improvement in responsivity was reached at $h≃h_{\mathrm{QD}}$, or when the air holes reached the depth of the buried QD layers. At small *h* values, the photocurrent enhancement mainly came from vertical resonance with a small contribution from the lateral modes, which was consistent with the measured result for the reflection spectra. At h>100 nm, the multiple absorption enhancement could not be explained by the anti-reflection property alone, and the waveguiding effects began to dominate. When the holes were formed by etching through the entire Ge/Si layer, the number of QDs in the active region (and hence the absorption) was reduced. Aside from that, at large *h* values, the surface recombination of the photogenerated carriers was increased due to the increase in the hole surface area, which also led to a decrease in responsivity.

## 4. Conclusions

In summary, we introduced light-trapping microstructures into vertical Ge/Si pin photodiodes with self-assembled Ge quantum dots grown on an SOI substrate. The hole arrays with different depths ranging from 100 nm to 550 nm served as the light-trapping structures by promoting laterally propogating modes. We addressed the trade-off between the amount of absorbing QD layers remaining after etching the holes and the enhanced light intensity in the region of the QDs due to the generation of guiding collective modes. Compared with a flat detector without holes, a photocurrent enhancement of about 50 times at 1.7 μm was achieved when the thickness of the photon-trapping structure reached the depth of the buried QD layers. The independence of the optical response from the polarization of the incident radiation was also established.

## Figures and Tables

**Figure 1 nanomaterials-12-02993-f001:**
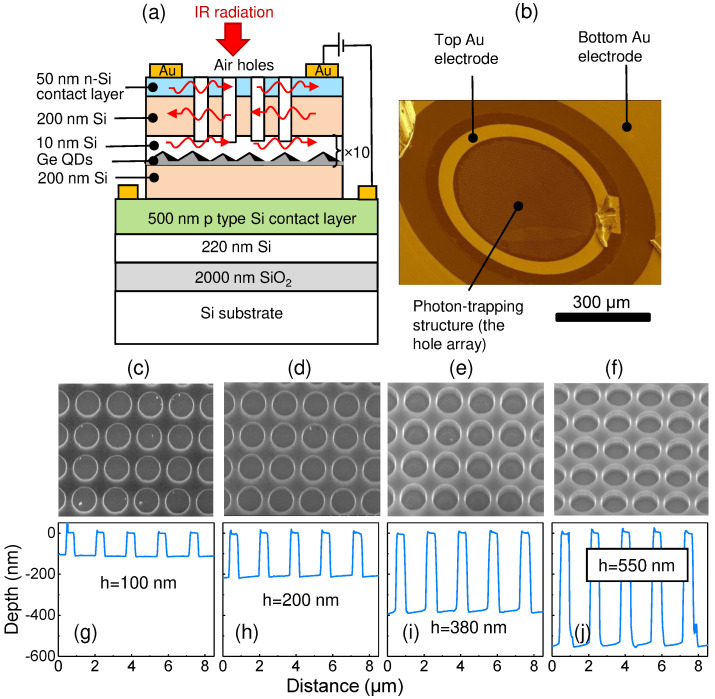
(**a**) Schematic of the pin QDIP structure on an SOI wafer with holes etched through the top n- and i-layers. (**b**) Optical image of a Ge/Si QDIP with a photon-trapping structure (top view). (**c**–**f**) Zoomed-in SEM images and (**g**–**j**) AFM profiles of the square lattice of circular holes with different hole depths *h*. The hole diameter is 1300 nm, and the lattice periodicity is 1700 nm. The hole depth is (**c**,**g**) 100 nm, (**d**,**h**) 200 nm, (**e**,**i**) 380 nm, and (**f**,**j**) 550 nm.

**Figure 2 nanomaterials-12-02993-f002:**
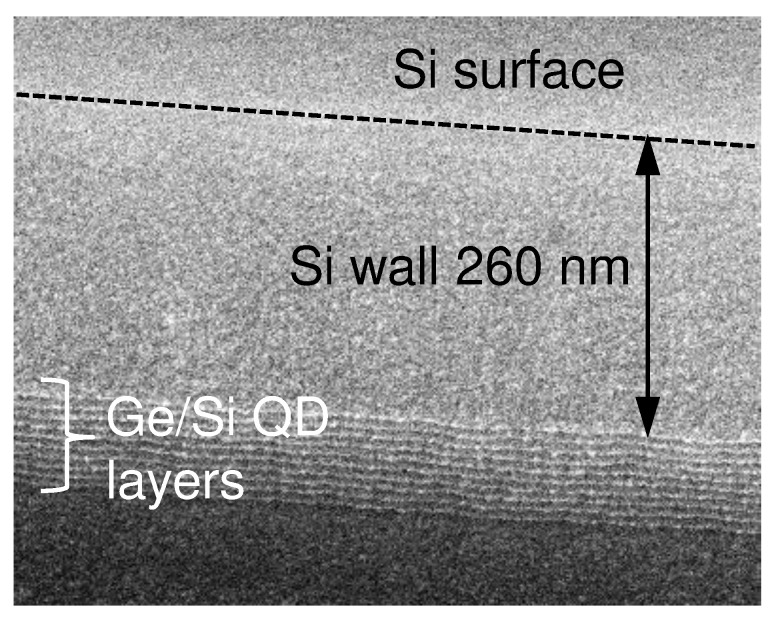
SEM micrograph of the active region of a Ge/Si QDIP with a 550-nm hole depth.

**Figure 3 nanomaterials-12-02993-f003:**
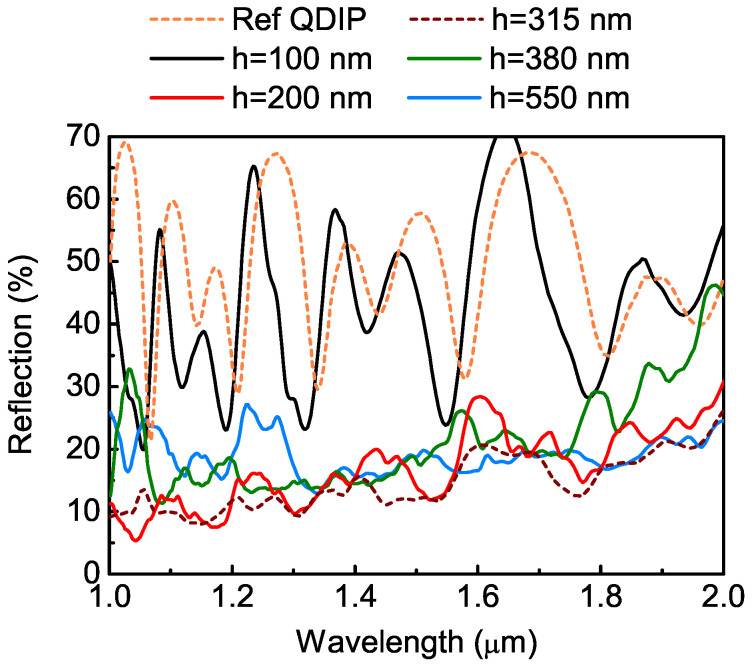
Reflection spectra of the Ge/Si QDIPs with and without photon-trapping microstructure from 1 to 2 μm. The hole depths are 100, 200, 315, 380, and 550 nm.

**Figure 4 nanomaterials-12-02993-f004:**
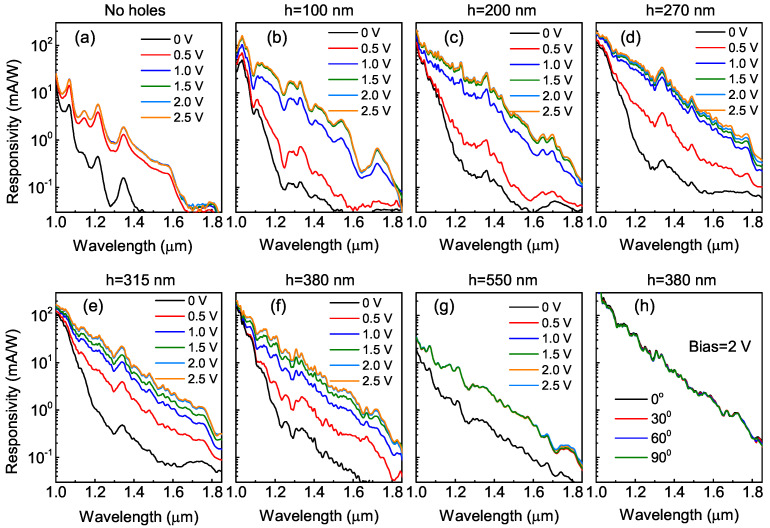
Responsivity spectra of the Ge/Si QDIPs (**a**) without and (**b**–**h**) with photon-trapping hole arrays. The hole depths were (**b**) 100, (**c**) 200, (**d**) 270, (**e**) 315, (**f**) 380, and (**g**) 550 nm. (**h**) Responsivity spectra of the light-trapping device with *h* = 380 nm at different polarization angles. The reverse bias was 2 V.

**Figure 5 nanomaterials-12-02993-f005:**
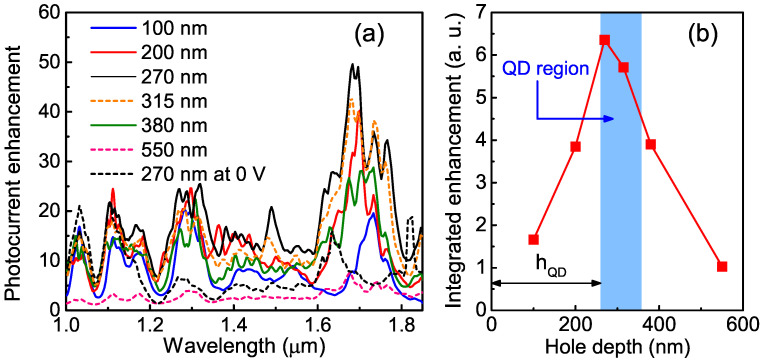
(**a**) Photocurrent enhancement spectra at 2.5 V of photon-trapping devices with different hole depths relative to the control detector. The photocurrent enhancement of the 270-nm hole device at a 0-V bias is shown by the black broken line. (**b**) Enhancement factor integrated over wavelengths in the range of 1.6–1.8 μm as a function of the hole depth *h*. The active region of the Ge quantum dots is inside the blue area. $h_{\mathrm{QD}}$ is the distance from the surface to the QD active region’s location. The maximum enhancement was observed when $h≃h_{\mathrm{QD}}$.

## Data Availability

Data are contained within the article.

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
