# Peer review of "Light-Trapping-Enhanced Photodetection in Ge/Si Quantum Dot Photodiodes Containing Microhole Arrays with Different Hole Depths"

_nanomaterials, 2022, doi:10.3390/nano12172993_

Round 1

Reviewer 1 Report

The results obtained by authors, i.e. the great enhancement of photocurrent in the 1600 – 1800 nm are of very high importance for developing photodiodes based on Ge and Si operating at room temperature.

                I only have minor comments mainly related to the measurement methodology, influence of Ge QDs morphology and multiple photocurrent peaks:

- Please add details on: how you measured the responsivity (you used powermeter?); if you used long-wave pass filters and what cut-on wavelengths, did you consider second-order blocking; add type and power of lamp source, also lamp spectral intensity if possible.

- Please explain the multiple photocurrent peaks (origin if possible): 1350 nm peak of sample without holes (Fig. 4a) seems to be also measured on samples with holes; Figure 4a: there is a shoulder at 1600 nm that is seems to be reproduced also for Figs. 4b,c, and for 0 V curves. What is the explanation for obtaining cut-off wavelength of 1800 nm, what is responsible for such extension? The authors state “broad photoresponse in the NIR spectral range up to 1.8 μm is due to interband transitions between the electron states in the conduction band and the hole states bound inside Ge QD”. Ge QDs are strained? 

- Please explain 550 nm hole behavior compared to no hole case and other holes cases for spectral photocurrent (Fig. 4).

- Please detail explanation on Fig. 4 for spectral photocurrent in function of hole size.

- Please add in caption of Figure 5a the 2.5 V bias voltage

- What is the maximum reverse voltage that can be applied?

- What is the photocurrent enhancement in photovoltaic regime (0 V bias)?

- Can you provide spectral photocurrent also in VIS, for example from 400 to 1800 nm or from 800 to 1800 nm for both photovoltaic and photoconduction regime?

- Is there any influence from Si by surface photovoltage?

- Did you perform measurements of photocurrent spectra for different temperatures (cooled devices)? It would be interesting to know at least what is the value of responsivity.

- Can you provide dark IV characteristics and photocurrent – voltage characteristics?

- Related to Ge QD pyramids, why small 0.9 nm thickness ? – please explain. They are connected to each other? What is the lateral size of Ge QDs? Ge QDs are strained? 

- The authors proposed 10 pairs of Ge QDs/Si. Did you check if there is any responsivity enhancement by varying the number of pairs? What about the density of Ge QDs?

Author Response

Referee: 1

Please add details on: how you measured the responsivity (you used powermeter?); if you used long-wave pass filters and what cut-on wavelengths, did you consider second-order blocking; add type and power of lamp source, also lamp spectral intensity if possible.

Authors:

The normal-incidence photoresponse was obtained using a Bruker Vertex 70 Fourier spectrometer with a spectral resolution of 30 cm-1 along with a SR570 low noise current preamplifier. A halogen lamp was used as a source of radiation. The photocurrent spectra were calibrated with a deuterated L-alanine doped triglycine sulfate detector. The responsivity was  determined by illuminating the samples with the InGaAsP LEDs (Roithner Laser Technik), which were emitting at 1.3 and 1.55 µm. Similar procedure has been described previously in [4]. This text is added in the revised paper (lines 106-108).

Referee: 1

Please explain the multiple photocurrent peaks (origin if possible): 1350 nm peak of sample without holes (Fig. 4a) seems to be also measured on samples with holes; Figure 4a: there is a shoulder at 1600 nm that is seems to be reproduced also for Figs. 4b,c, and for 0 V curves.

What is the explanation for obtaining cut-off wavelength of 1800 nm, what is responsible for such extension? The authors state “broad photoresponse in the NIR spectral range up to 1.8 μm is due to interband transitions between the electron states in the conduction band and the hole states bound inside Ge QD”. Ge QDs are strained? 

Authors:

Employment  of a SOI substrate makes it possible for the vertical Fabry-Perot resonances between SOI substrate and top surface due to the large contrast in the refractive index between SiO2  and Si layers. Light propogating in the Ge/Si active region can experience a strong reflection at the buried oxide/Si and Si/air interfaces, creating a vertical resonant cavity which results in development of a standing wave pattern in all samples [10,29]. Constructive optical interference of incident and reflected wave leads to photocurrent  maxima, destructive interference results in PC minima. It is the origin of the multiple photocurrent peaks. However, in light-trapping devices, photons are coupled to waveguiding planar modes and the vertical resonances become less evident.

The cut-off wavelength of 1800 nm corresponds to the ground state intraband transition in Ge/Si QDs. Ge/Si QDs embedded in Si(001) have a type-II band alignment. Here, the charge carries of different signs are separated by the heterointerface. In the valence band, the holes are confined inside the Ge QD, while the electrons are in the conduction band of Si. The calculated and experimental band diagrams for different Ge dot sizes have been published in many papers [see, Ref.4]. We believe that there is no need to repeat the details in present work.

Referee:1  Ge QDs are strained? 

Authors: Ge QDs are pseudomorphically strained. The average Ge content of 88% and the in-plane strain of -0.036 in Ge nanoclusters were determined from Raman scattering experiments [4]. This information is added in lines 79, 80 of the revised text.

Referee: 1

 Please explain 550 nm hole behavior compared to no hole case and other holes cases for spectral photocurrent (Fig. 4).

Please detail explanation on Fig. 4 for spectral photocurrent in function of hole size.

Authors:

The origin of the features of the photocurrent spectra is similar to the features of the reflection spectra, which are described in section Results and Discussion, first paragraph. In comparison between control and light-trapping devices, the control detector has sharp resonances, suggesting a strong vertical resonance between SOI substrate and top surface. Light-trapping detectors have much weaker vertical resonances, suggesting that photons are coupled to lateral modes. This effect becomes more pronounced with increasing the hole depth. Referee: 1Please add in caption of Figure 5a the 2.5 V bias voltage. 

Authors:

The 2.5 V bias voltage is added in caption of Figure 5a. Referee:1

What is the maximum reverse voltage that can be applied?

Authors:

The maximum reverse bias of about 3 V can be applied. However, at reverse bias larger than 1 V, little change in the responsivity is observed. Referee:1

What is the photocurrent enhancement in photovoltaic regime (0 V bias)?

Authors:

We add the enhancement spectrum of the 270-nm hole device at 0 V bias in Figure5a. The following text is added in the revised paper (lines 143-147): We found that photocurrent enhancement in photovoltaic regime is small, it increases with increasing reverse bias and ceases to depend on voltage as the reverse bias voltage reaches »1V. This observation can be explained by electrically-controlled overlap between the depletion layer and the light-trapping region [30]. Referee:1

Can you provide spectral photocurrent also in VIS, for example from 400 to 1800 nm or from 800 to 1800 nm for both photovoltaic and photoconduction regime?

Authors:

Unfortunately, we do not have the technical capability to carry out optical measurements at wavelengths shorter 1 um. However, VIS experiments are well described in [10, 29]. Referee:1Is there any influence from Si by surface photovoltage? 

Authors:

The influence of surface photovoltage in our samples was not observed. Referee:1

Did you perform measurements of photocurrent spectra for different temperatures (cooled devices)? It would be interesting to know at least what is the value of responsivity.

Authors:

The aim of our present and previous works is to demonstrate the GeSi-based quantum dot photodiodes with improved efficiency operating at room temperature. We did not perform measurements at low temperatures. Perhaps it will be done in the future papers.

Referee:1Can you provide dark  I–V characteristics and photocurrent-voltage characteristics? Authors:Dark current-voltage characteristics of the photodiodes described in the present paper are similar to that described in [26]. Referee: 1

Related to Ge QD pyramids, why small 0.9 nm thickness ? – please explain. They are connected to each other? What is the lateral size of Ge QDs? Ge QDs are strained? 

Authors:

The details of Ge/Si QD growth were described in [4] and in references therein. QDs are formed by self-assembling in the Stranski-Krastanov growth mode. QDs have a hut-like shape with a lateral size of 9.4 \pm 3.2 nm and a height of about 1 nm. The surface density of QDs is 5.2x 1011 cm-2.  The average Ge content of 88% and the in-plane strain of  0.036 in Ge nanoclusters were determined from Raman scattering experiments. This information is added in the revised text (lines 78-81).  If the nominal Ge coverage exceeds about 1.2-1.5 nm, the strain is relaxed and dislocation-rich structure appears.

STM images of Ge/Si QDs can be found in [28].

Referee: 1

The authors proposed 10 pairs of Ge QDs/Si. Did you check if there is any responsivity enhancement by varying the number of pairs? What about the density of Ge QDs?

Authors:

Certainly responsivity increases with increasing the number of QD layers. However, the total number of QD layers that can be incorporated in a QD infrared photodetector is limited by the accumulation of strain and the strain induced defects and dislocations, resulting in a thin active QD absorption region. This limits the total responsivity of QD detectors.

The surface density of QDs is 5.2x 1011 cm-2.

Referee: 1

Please add details on: how you measured the responsivity (you used powermeter?); if you used long-wave pass filters and what cut-on wavelengths, did you consider second-order blocking; add type and power of lamp source, also lamp spectral intensity if possible.

Authors:

The normal-incidence photoresponse was obtained using a Bruker Vertex 70 Fourier spectrometer with a spectral resolution of 30 cm-1 along with a SR570 low noise current preamplifier. A halogen lamp was used as a source of radiation. The photocurrent spectra were calibrated with a deuterated L-alanine doped triglycine sulfate detector. The responsivity was  determined by illuminating the samples with the InGaAsP LEDs (Roithner Laser Technik), which were emitting at 1.3 and 1.55 µm. Similar procedure has been described previously in [4]. This text is added in the revised paper (lines 106-108).

Referee: 1

Please explain the multiple photocurrent peaks (origin if possible): 1350 nm peak of sample without holes (Fig. 4a) seems to be also measured on samples with holes; Figure 4a: there is a shoulder at 1600 nm that is seems to be reproduced also for Figs. 4b,c, and for 0 V curves.

What is the explanation for obtaining cut-off wavelength of 1800 nm, what is responsible for such extension? The authors state “broad photoresponse in the NIR spectral range up to 1.8 μm is due to interband transitions between the electron states in the conduction band and the hole states bound inside Ge QD”. Ge QDs are strained? 

Authors:

Employment  of a SOI substrate makes it possible for the vertical Fabry-Perot resonances between SOI substrate and top surface due to the large contrast in the refractive index between SiO2  and Si layers. Light propogating in the Ge/Si active region can experience a strong reflection at the buried oxide/Si and Si/air interfaces, creating a vertical resonant cavity which results in development of a standing wave pattern in all samples [10,29]. Constructive optical interference of incident and reflected wave leads to photocurrent  maxima, destructive interference results in PC minima. It is the origin of the multiple photocurrent peaks. However, in light-trapping devices, photons are coupled to waveguiding planar modes and the vertical resonances become less evident.

The cut-off wavelength of 1800 nm corresponds to the ground state intraband transition in Ge/Si QDs. Ge/Si QDs embedded in Si(001) have a type-II band alignment. Here, the charge carries of different signs are separated by the heterointerface. In the valence band, the holes are confined inside the Ge QD, while the electrons are in the conduction band of Si. The calculated and experimental band diagrams for different Ge dot sizes have been published in many papers [see, Ref.4]. We believe that there is no need to repeat the details in present work.

Referee:1  Ge QDs are strained? 

Authors: Ge QDs are pseudomorphically strained. The average Ge content of 88% and the in-plane strain of -0.036 in Ge nanoclusters were determined from Raman scattering experiments [4]. This information is added in lines 79, 80 of the revised text.

Referee: 1

 Please explain 550 nm hole behavior compared to no hole case and other holes cases for spectral photocurrent (Fig. 4).

Please detail explanation on Fig. 4 for spectral photocurrent in function of hole size.

Authors:

The origin of the features of the photocurrent spectra is similar to the features of the reflection spectra, which are described in section Results and Discussion, first paragraph. In comparison between control and light-trapping devices, the control detector has sharp resonances, suggesting a strong vertical resonance between SOI substrate and top surface. Light-trapping detectors have much weaker vertical resonances, suggesting that photons are coupled to lateral modes. This effect becomes more pronounced with increasing the hole depth. Referee: 1Please add in caption of Figure 5a the 2.5 V bias voltage. 

Authors:

The 2.5 V bias voltage is added in caption of Figure 5a. Referee:1

What is the maximum reverse voltage that can be applied?

Authors:

The maximum reverse bias of about 3 V can be applied. However, at reverse bias larger than 1 V, little change in the responsivity is observed. Referee:1

What is the photocurrent enhancement in photovoltaic regime (0 V bias)?

Authors:

We add the enhancement spectrum of the 270-nm hole device at 0 V bias in Figure5a. The following text is added in the revised paper (lines 143-147): We found that photocurrent enhancement in photovoltaic regime is small, it increases with increasing reverse bias and ceases to depend on voltage as the reverse bias voltage reaches »1V. This observation can be explained by electrically-controlled overlap between the depletion layer and the light-trapping region [30]. Referee:1

Can you provide spectral photocurrent also in VIS, for example from 400 to 1800 nm or from 800 to 1800 nm for both photovoltaic and photoconduction regime?

Authors:

Unfortunately, we do not have the technical capability to carry out optical measurements at wavelengths shorter 1 um. However, VIS experiments are well described in [10, 29]. Referee:1Is there any influence from Si by surface photovoltage? 

Authors:

The influence of surface photovoltage in our samples was not observed. Referee:1

Did you perform measurements of photocurrent spectra for different temperatures (cooled devices)? It would be interesting to know at least what is the value of responsivity.

Authors:

The aim of our present and previous works is to demonstrate the GeSi-based quantum dot photodiodes with improved efficiency operating at room temperature. We did not perform measurements at low temperatures. Perhaps it will be done in the future papers.

Referee:1Can you provide dark  I–V characteristics and photocurrent-voltage characteristics? Authors:Dark current-voltage characteristics of the photodiodes described in the present paper are similar to that described in [26]. Referee: 1

Related to Ge QD pyramids, why small 0.9 nm thickness ? – please explain. They are connected to each other? What is the lateral size of Ge QDs? Ge QDs are strained? 

Authors:

The details of Ge/Si QD growth were described in [4] and in references therein. QDs are formed by self-assembling in the Stranski-Krastanov growth mode. QDs have a hut-like shape with a lateral size of 9.4 \pm 3.2 nm and a height of about 1 nm. The surface density of QDs is 5.2x 1011 cm-2.  The average Ge content of 88% and the in-plane strain of  0.036 in Ge nanoclusters were determined from Raman scattering experiments. This information is added in the revised text (lines 78-81).  If the nominal Ge coverage exceeds about 1.2-1.5 nm, the strain is relaxed and dislocation-rich structure appears.

STM images of Ge/Si QDs can be found in [28].

Referee: 1

The authors proposed 10 pairs of Ge QDs/Si. Did you check if there is any responsivity enhancement by varying the number of pairs? What about the density of Ge QDs?

Authors:

Certainly responsivity increases with increasing the number of QD layers. However, the total number of QD layers that can be incorporated in a QD infrared photodetector is limited by the accumulation of strain and the strain induced defects and dislocations, resulting in a thin active QD absorption region. This limits the total responsivity of QD detectors.

The surface density of QDs is 5.2x 1011 cm-2.

Referee: 1

Please add details on: how you measured the responsivity (you used powermeter?); if you used long-wave pass filters and what cut-on wavelengths, did you consider second-order blocking; add type and power of lamp source, also lamp spectral intensity if possible.

Authors:

The normal-incidence photoresponse was obtained using a Bruker Vertex 70 Fourier spectrometer with a spectral resolution of 30 cm-1 along with a SR570 low noise current preamplifier. A halogen lamp was used as a source of radiation. The photocurrent spectra were calibrated with a deuterated L-alanine doped triglycine sulfate detector. The responsivity was  determined by illuminating the samples with the InGaAsP LEDs (Roithner Laser Technik), which were emitting at 1.3 and 1.55 µm. Similar procedure has been described previously in [4]. This text is added in the revised paper (lines 106-108).

Referee: 1

Please explain the multiple photocurrent peaks (origin if possible): 1350 nm peak of sample without holes (Fig. 4a) seems to be also measured on samples with holes; Figure 4a: there is a shoulder at 1600 nm that is seems to be reproduced also for Figs. 4b,c, and for 0 V curves.

What is the explanation for obtaining cut-off wavelength of 1800 nm, what is responsible for such extension? The authors state “broad photoresponse in the NIR spectral range up to 1.8 μm is due to interband transitions between the electron states in the conduction band and the hole states bound inside Ge QD”. Ge QDs are strained? 

Authors:

Employment  of a SOI substrate makes it possible for the vertical Fabry-Perot resonances between SOI substrate and top surface due to the large contrast in the refractive index between SiO2  and Si layers. Light propogating in the Ge/Si active region can experience a strong reflection at the buried oxide/Si and Si/air interfaces, creating a vertical resonant cavity which results in development of a standing wave pattern in all samples [10,29]. Constructive optical interference of incident and reflected wave leads to photocurrent  maxima, destructive interference results in PC minima. It is the origin of the multiple photocurrent peaks. However, in light-trapping devices, photons are coupled to waveguiding planar modes and the vertical resonances become less evident.

The cut-off wavelength of 1800 nm corresponds to the ground state intraband transition in Ge/Si QDs. Ge/Si QDs embedded in Si(001) have a type-II band alignment. Here, the charge carries of different signs are separated by the heterointerface. In the valence band, the holes are confined inside the Ge QD, while the electrons are in the conduction band of Si. The calculated and experimental band diagrams for different Ge dot sizes have been published in many papers [see, Ref.4]. We believe that there is no need to repeat the details in present work.

Referee:1  Ge QDs are strained? 

Authors: Ge QDs are pseudomorphically strained. The average Ge content of 88% and the in-plane strain of -0.036 in Ge nanoclusters were determined from Raman scattering experiments [4]. This information is added in lines 79, 80 of the revised text.

Referee: 1

 Please explain 550 nm hole behavior compared to no hole case and other holes cases for spectral photocurrent (Fig. 4).

Please detail explanation on Fig. 4 for spectral photocurrent in function of hole size.

Authors:

The origin of the features of the photocurrent spectra is similar to the features of the reflection spectra, which are described in section Results and Discussion, first paragraph. In comparison between control and light-trapping devices, the control detector has sharp resonances, suggesting a strong vertical resonance between SOI substrate and top surface. Light-trapping detectors have much weaker vertical resonances, suggesting that photons are coupled to lateral modes. This effect becomes more pronounced with increasing the hole depth. Referee: 1Please add in caption of Figure 5a the 2.5 V bias voltage. 

Authors:

The 2.5 V bias voltage is added in caption of Figure 5a. Referee:1

What is the maximum reverse voltage that can be applied?

Authors:

The maximum reverse bias of about 3 V can be applied. However, at reverse bias larger than 1 V, little change in the responsivity is observed. Referee:1

What is the photocurrent enhancement in photovoltaic regime (0 V bias)?

Authors:

We add the enhancement spectrum of the 270-nm hole device at 0 V bias in Figure5a. The following text is added in the revised paper (lines 143-147): We found that photocurrent enhancement in photovoltaic regime is small, it increases with increasing reverse bias and ceases to depend on voltage as the reverse bias voltage reaches »1V. This observation can be explained by electrically-controlled overlap between the depletion layer and the light-trapping region [30]. Referee:1

Can you provide spectral photocurrent also in VIS, for example from 400 to 1800 nm or from 800 to 1800 nm for both photovoltaic and photoconduction regime?

Authors:

Unfortunately, we do not have the technical capability to carry out optical measurements at wavelengths shorter 1 um. However, VIS experiments are well described in [10, 29]. Referee:1Is there any influence from Si by surface photovoltage? 

Authors:

The influence of surface photovoltage in our samples was not observed. Referee:1

Did you perform measurements of photocurrent spectra for different temperatures (cooled devices)? It would be interesting to know at least what is the value of responsivity.

Authors:

The aim of our present and previous works is to demonstrate the GeSi-based quantum dot photodiodes with improved efficiency operating at room temperature. We did not perform measurements at low temperatures. Perhaps it will be done in the future papers.

Referee:1Can you provide dark  I–V characteristics and photocurrent-voltage characteristics? Authors:Dark current-voltage characteristics of the photodiodes described in the present paper are similar to that described in [26]. Referee: 1

Related to Ge QD pyramids, why small 0.9 nm thickness ? – please explain. They are connected to each other? What is the lateral size of Ge QDs? Ge QDs are strained? 

Authors:

The details of Ge/Si QD growth were described in [4] and in references therein. QDs are formed by self-assembling in the Stranski-Krastanov growth mode. QDs have a hut-like shape with a lateral size of 9.4 \pm 3.2 nm and a height of about 1 nm. The surface density of QDs is 5.2x 1011 cm-2.  The average Ge content of 88% and the in-plane strain of  0.036 in Ge nanoclusters were determined from Raman scattering experiments. This information is added in the revised text (lines 78-81).  If the nominal Ge coverage exceeds about 1.2-1.5 nm, the strain is relaxed and dislocation-rich structure appears.

STM images of Ge/Si QDs can be found in [28].

Referee: 1

The authors proposed 10 pairs of Ge QDs/Si. Did you check if there is any responsivity enhancement by varying the number of pairs? What about the density of Ge QDs?

Authors:

Certainly responsivity increases with increasing the number of QD layers. However, the total number of QD layers that can be incorporated in a QD infrared photodetector is limited by the accumulation of strain and the strain induced defects and dislocations, resulting in a thin active QD absorption region. This limits the total responsivity of QD detectors.

The surface density of QDs is 5.2x 1011 cm-2.

Referee: 1

Please add details on: how you measured the responsivity (you used powermeter?); if you used long-wave pass filters and what cut-on wavelengths, did you consider second-order blocking; add type and power of lamp source, also lamp spectral intensity if possible.

Authors:

The normal-incidence photoresponse was obtained using a Bruker Vertex 70 Fourier spectrometer with a spectral resolution of 30 cm-1 along with a SR570 low noise current preamplifier. A halogen lamp was used as a source of radiation. The photocurrent spectra were calibrated with a deuterated L-alanine doped triglycine sulfate detector. The responsivity was  determined by illuminating the samples with the InGaAsP LEDs (Roithner Laser Technik), which were emitting at 1.3 and 1.55 µm. Similar procedure has been described previously in [4]. This text is added in the revised paper (lines 106-108).

Referee: 1

Please explain the multiple photocurrent peaks (origin if possible): 1350 nm peak of sample without holes (Fig. 4a) seems to be also measured on samples with holes; Figure 4a: there is a shoulder at 1600 nm that is seems to be reproduced also for Figs. 4b,c, and for 0 V curves.

What is the explanation for obtaining cut-off wavelength of 1800 nm, what is responsible for such extension? The authors state “broad photoresponse in the NIR spectral range up to 1.8 μm is due to interband transitions between the electron states in the conduction band and the hole states bound inside Ge QD”. Ge QDs are strained? 

Authors:

Employment  of a SOI substrate makes it possible for the vertical Fabry-Perot resonances between SOI substrate and top surface due to the large contrast in the refractive index between SiO2  and Si layers. Light propogating in the Ge/Si active region can experience a strong reflection at the buried oxide/Si and Si/air interfaces, creating a vertical resonant cavity which results in development of a standing wave pattern in all samples [10,29]. Constructive optical interference of incident and reflected wave leads to photocurrent  maxima, destructive interference results in PC minima. It is the origin of the multiple photocurrent peaks. However, in light-trapping devices, photons are coupled to waveguiding planar modes and the vertical resonances become less evident.

The cut-off wavelength of 1800 nm corresponds to the ground state intraband transition in Ge/Si QDs. Ge/Si QDs embedded in Si(001) have a type-II band alignment. Here, the charge carries of different signs are separated by the heterointerface. In the valence band, the holes are confined inside the Ge QD, while the electrons are in the conduction band of Si. The calculated and experimental band diagrams for different Ge dot sizes have been published in many papers [see, Ref.4]. We believe that there is no need to repeat the details in present work.

Referee:1  Ge QDs are strained? 

Authors: Ge QDs are pseudomorphically strained. The average Ge content of 88% and the in-plane strain of -0.036 in Ge nanoclusters were determined from Raman scattering experiments [4]. This information is added in lines 79, 80 of the revised text.

Referee: 1

 Please explain 550 nm hole behavior compared to no hole case and other holes cases for spectral photocurrent (Fig. 4).

Please detail explanation on Fig. 4 for spectral photocurrent in function of hole size.

Authors:

The origin of the features of the photocurrent spectra is similar to the features of the reflection spectra, which are described in section Results and Discussion, first paragraph. In comparison between control and light-trapping devices, the control detector has sharp resonances, suggesting a strong vertical resonance between SOI substrate and top surface. Light-trapping detectors have much weaker vertical resonances, suggesting that photons are coupled to lateral modes. This effect becomes more pronounced with increasing the hole depth. Referee: 1Please add in caption of Figure 5a the 2.5 V bias voltage. 

Authors:

The 2.5 V bias voltage is added in caption of Figure 5a. Referee:1

What is the maximum reverse voltage that can be applied?

Authors:

The maximum reverse bias of about 3 V can be applied. However, at reverse bias larger than 1 V, little change in the responsivity is observed. Referee:1

What is the photocurrent enhancement in photovoltaic regime (0 V bias)?

Authors:

We add the enhancement spectrum of the 270-nm hole device at 0 V bias in Figure5a. The following text is added in the revised paper (lines 143-147): We found that photocurrent enhancement in photovoltaic regime is small, it increases with increasing reverse bias and ceases to depend on voltage as the reverse bias voltage reaches »1V. This observation can be explained by electrically-controlled overlap between the depletion layer and the light-trapping region [30]. Referee:1

Can you provide spectral photocurrent also in VIS, for example from 400 to 1800 nm or from 800 to 1800 nm for both photovoltaic and photoconduction regime?

Authors:

Unfortunately, we do not have the technical capability to carry out optical measurements at wavelengths shorter 1 um. However, VIS experiments are well described in [10, 29]. Referee:1Is there any influence from Si by surface photovoltage? 

Authors:

The influence of surface photovoltage in our samples was not observed. Referee:1

Did you perform measurements of photocurrent spectra for different temperatures (cooled devices)? It would be interesting to know at least what is the value of responsivity.

Authors:

The aim of our present and previous works is to demonstrate the GeSi-based quantum dot photodiodes with improved efficiency operating at room temperature. We did not perform measurements at low temperatures. Perhaps it will be done in the future papers.

Referee:1Can you provide dark  I–V characteristics and photocurrent-voltage characteristics? Authors:Dark current-voltage characteristics of the photodiodes described in the present paper are similar to that described in [26]. Referee: 1

Related to Ge QD pyramids, why small 0.9 nm thickness ? – please explain. They are connected to each other? What is the lateral size of Ge QDs? Ge QDs are strained? 

Authors:

The details of Ge/Si QD growth were described in [4] and in references therein. QDs are formed by self-assembling in the Stranski-Krastanov growth mode. QDs have a hut-like shape with a lateral size of 9.4 \pm 3.2 nm and a height of about 1 nm. The surface density of QDs is 5.2x 1011 cm-2.  The average Ge content of 88% and the in-plane strain of  0.036 in Ge nanoclusters were determined from Raman scattering experiments. This information is added in the revised text (lines 78-81).  If the nominal Ge coverage exceeds about 1.2-1.5 nm, the strain is relaxed and dislocation-rich structure appears.

STM images of Ge/Si QDs can be found in [28].

Referee: 1

The authors proposed 10 pairs of Ge QDs/Si. Did you check if there is any responsivity enhancement by varying the number of pairs? What about the density of Ge QDs?

Authors:

Certainly responsivity increases with increasing the number of QD layers. However, the total number of QD layers that can be incorporated in a QD infrared photodetector is limited by the accumulation of strain and the strain induced defects and dislocations, resulting in a thin active QD absorption region. This limits the total responsivity of QD detectors.

The surface density of QDs is 5.2x 1011 cm-2.

Referee: 1

Please add details on: how you measured the responsivity (you used powermeter?); if you used long-wave pass filters and what cut-on wavelengths, did you consider second-order blocking; add type and power of lamp source, also lamp spectral intensity if possible.

Authors:

The normal-incidence photoresponse was obtained using a Bruker Vertex 70 Fourier spectrometer with a spectral resolution of 30 cm-1 along with a SR570 low noise current preamplifier. A halogen lamp was used as a source of radiation. The photocurrent spectra were calibrated with a deuterated L-alanine doped triglycine sulfate detector. The responsivity was  determined by illuminating the samples with the InGaAsP LEDs (Roithner Laser Technik), which were emitting at 1.3 and 1.55 µm. Similar procedure has been described previously in [4]. This text is added in the revised paper (lines 106-108).

Referee: 1

Please explain the multiple photocurrent peaks (origin if possible): 1350 nm peak of sample without holes (Fig. 4a) seems to be also measured on samples with holes; Figure 4a: there is a shoulder at 1600 nm that is seems to be reproduced also for Figs. 4b,c, and for 0 V curves.

What is the explanation for obtaining cut-off wavelength of 1800 nm, what is responsible for such extension? The authors state “broad photoresponse in the NIR spectral range up to 1.8 μm is due to interband transitions between the electron states in the conduction band and the hole states bound inside Ge QD”. Ge QDs are strained? 

Authors:

Employment  of a SOI substrate makes it possible for the vertical Fabry-Perot resonances between SOI substrate and top surface due to the large contrast in the refractive index between SiO2  and Si layers. Light propogating in the Ge/Si active region can experience a strong reflection at the buried oxide/Si and Si/air interfaces, creating a vertical resonant cavity which results in development of a standing wave pattern in all samples [10,29]. Constructive optical interference of incident and reflected wave leads to photocurrent  maxima, destructive interference results in PC minima. It is the origin of the multiple photocurrent peaks. However, in light-trapping devices, photons are coupled to waveguiding planar modes and the vertical resonances become less evident.

The cut-off wavelength of 1800 nm corresponds to the ground state intraband transition in Ge/Si QDs. Ge/Si QDs embedded in Si(001) have a type-II band alignment. Here, the charge carries of different signs are separated by the heterointerface. In the valence band, the holes are confined inside the Ge QD, while the electrons are in the conduction band of Si. The calculated and experimental band diagrams for different Ge dot sizes have been published in many papers [see, Ref.4]. We believe that there is no need to repeat the details in present work.

Referee:1  Ge QDs are strained? 

Authors: Ge QDs are pseudomorphically strained. The average Ge content of 88% and the in-plane strain of -0.036 in Ge nanoclusters were determined from Raman scattering experiments [4]. This information is added in lines 79, 80 of the revised text.

Referee: 1

 Please explain 550 nm hole behavior compared to no hole case and other holes cases for spectral photocurrent (Fig. 4).

Please detail explanation on Fig. 4 for spectral photocurrent in function of hole size.

Authors:

The origin of the features of the photocurrent spectra is similar to the features of the reflection spectra, which are described in section Results and Discussion, first paragraph. In comparison between control and light-trapping devices, the control detector has sharp resonances, suggesting a strong vertical resonance between SOI substrate and top surface. Light-trapping detectors have much weaker vertical resonances, suggesting that photons are coupled to lateral modes. This effect becomes more pronounced with increasing the hole depth. Referee: 1Please add in caption of Figure 5a the 2.5 V bias voltage. 

Authors:

The 2.5 V bias voltage is added in caption of Figure 5a. Referee:1

What is the maximum reverse voltage that can be applied?

Authors:

The maximum reverse bias of about 3 V can be applied. However, at reverse bias larger than 1 V, little change in the responsivity is observed. Referee:1

What is the photocurrent enhancement in photovoltaic regime (0 V bias)?

Authors:

We add the enhancement spectrum of the 270-nm hole device at 0 V bias in Figure5a. The following text is added in the revised paper (lines 143-147): We found that photocurrent enhancement in photovoltaic regime is small, it increases with increasing reverse bias and ceases to depend on voltage as the reverse bias voltage reaches »1V. This observation can be explained by electrically-controlled overlap between the depletion layer and the light-trapping region [30]. Referee:1

Can you provide spectral photocurrent also in VIS, for example from 400 to 1800 nm or from 800 to 1800 nm for both photovoltaic and photoconduction regime?

Authors:

Unfortunately, we do not have the technical capability to carry out optical measurements at wavelengths shorter 1 um. However, VIS experiments are well described in [10, 29]. Referee:1Is there any influence from Si by surface photovoltage? 

Authors:

The influence of surface photovoltage in our samples was not observed. Referee:1

Did you perform measurements of photocurrent spectra for different temperatures (cooled devices)? It would be interesting to know at least what is the value of responsivity.

Authors:

The aim of our present and previous works is to demonstrate the GeSi-based quantum dot photodiodes with improved efficiency operating at room temperature. We did not perform measurements at low temperatures. Perhaps it will be done in the future papers.

Referee:1Can you provide dark  I–V characteristics and photocurrent-voltage characteristics? Authors:Dark current-voltage characteristics of the photodiodes described in the present paper are similar to that described in [26]. Referee: 1

Related to Ge QD pyramids, why small 0.9 nm thickness ? – please explain. They are connected to each other? What is the lateral size of Ge QDs? Ge QDs are strained? 

Authors:

The details of Ge/Si QD growth were described in [4] and in references therein. QDs are formed by self-assembling in the Stranski-Krastanov growth mode. QDs have a hut-like shape with a lateral size of 9.4 \pm 3.2 nm and a height of about 1 nm. The surface density of QDs is 5.2x 1011 cm-2.  The average Ge content of 88% and the in-plane strain of  0.036 in Ge nanoclusters were determined from Raman scattering experiments. This information is added in the revised text (lines 78-81).  If the nominal Ge coverage exceeds about 1.2-1.5 nm, the strain is relaxed and dislocation-rich structure appears.

STM images of Ge/Si QDs can be found in [28].

Referee: 1

The authors proposed 10 pairs of Ge QDs/Si. Did you check if there is any responsivity enhancement by varying the number of pairs? What about the density of Ge QDs?

Authors:

Certainly responsivity increases with increasing the number of QD layers. However, the total number of QD layers that can be incorporated in a QD infrared photodetector is limited by the accumulation of strain and the strain induced defects and dislocations, resulting in a thin active QD absorption region. This limits the total responsivity of QD detectors.

The surface density of QDs is 5.2x 1011 cm-2.

Reviewer 2 Report

The author showed experimental results of photodetection effects in Ge/Si quantum dots with different densities versus infrared wavelengths (1-1.8um). The understanding of the quantum dot/plasmonic effect is well established from previous literature. This manuscript does not demonstrate/ explain well why these results are important/different compared to previous literature. I suggest that minor revision/restructure is required for the manuscript to be accepted. 

Please refer to similar papers from Vladimer Shaleev, Emmanuel Lhuillier, etc. 

For example: 

Livache, C, Martinez B, Goubet, N. et al A colloidal quantum dot infrared photodetector and its use for intraband detection. Nat Commun 10, 2125 (2019) 

Liu, H., Kang Y., et al. High Photon Absorptivity of Quantum Dot Infrared Photedetectors Achieved by the Surface Plasmon Effect of Metal Nanohole Array. Nanoscale Res Lett 15, 98 (2020) 

Wang, D., Koh, Y. R., Kudysev, Z. et al., Spatial and temporal nanoscale plasmonic heating quantified by thermoreflectance. Nano Lette. 19(6), 3796-3803 (2019)

Author Response

The author showed experimental results of photodetection effects in Ge/Si quantum dots with different densities versus infrared wavelengths (1-1.8um). The understanding of the quantum dot/plasmonic effect is well established from previous literature. This manuscript does not demonstrate/ explain well why these results are important/different compared to previous literature. I suggest that minor revision/restructure is required for the manuscript to be accepted. 

Please refer to similar papers from Vladimer Shaleev, Emmanuel Lhuillier, etc. 

For example: 

Livache, C, Martinez B, Goubet, N. et al A colloidal quantum dot infrared photodetector and its use for intraband detection. Nat Commun 10, 2125 (2019) 

Liu, H., Kang Y., et al. High Photon Absorptivity of Quantum Dot Infrared Photedetectors Achieved by the Surface Plasmon Effect of Metal Nanohole Array. Nanoscale Res Lett 15, 98 (2020) 

Wang, D., Koh, Y. R., Kudysev, Z. et al., Spatial and temporal nanoscale plasmonic heating quantified by thermoreflectance. Nano Lette. 19(6), 3796-3803 (2019)

Authors:

Different approaches to improve the QD photoresponse are currently under investigation. Of course, regular subwavelength gratings of holes in metallic films or arrays  of metallic nanoparticles on the surface of a semiconductor can increase the QD detector sensitivity due to effective surface light trapping, enhancement of local optical fields, and interaction with  thin active regions of optical devices. Disadvantages of metallic metasurfaces that allow conversion of the incident electromagnetic radiation into the surface plasmons are high ohmic losses in a metal and a small penetration depth of the plasmon field to a semiconductor, particularly for short wavelengths. By this reason, we do not discuss the quantum dot/plasmonic effect in our paper. Instead, we introduce the all-dielectric photon trapping structures, which offer a low-loss alternative to plasmon elements. Nonetheless, we refer to the papers indicated above by the author (see, [5, 8, 9]).

Round 2

Reviewer 2 Report

Authors had well responsed reviewer's comments. The revised manuscript can be accepted in Nanomaterials.